# The Impact of Mediterranean Dietary Patterns During Pregnancy on Maternal and Offspring Health

**DOI:** 10.3390/nu11051098

**Published:** 2019-05-17

**Authors:** Federica Amati, Sondus Hassounah, Alexandra Swaka

**Affiliations:** Department of Primary Care and Public Health, Imperial College London, London W6 8RP, UK; s.hassounah@imperial.ac.uk (S.H.); a.swaka@imperial.ac.uk (A.S.)

**Keywords:** maternal nutrition, Mediterranean diet, offspring health

## Abstract

(1) Background: Pregnancy outcomes for both mother and child are affected by many environmental factors. The importance of pregnancy for ‘early life programming’ is well established and maternal nutrition is an important factor contributing to a favourable environment for developing offspring. We aim to assess whether following a Mediterranean Diet during pregnancy is beneficial for maternal and offspring outcomes; (2) Methods: a systematic review was performed using standardized reporting guidelines with the National Heart Lung and Blood Iinstitute quality assessment tool for selection and extraction; (3) Results: results show that being on a Mediterranean Diet during pregnancy is associated with favourable outcomes for both maternal and offspring health, particularly for gestational diabetes in mothers and congenital defects in offspring (4) Conclusions: Following a Mediterranean dietary pattern during gestation is beneficial for the health of both the mother and offspring. Pregnant women and those trying to conceive should be advised to follow a Mediterranean Diet to potentially decrease, for example, the likelihood of atopy (OR 0.55) in the offspring and Gestational Diabetes Mellitus in the mother (OR 0.73).

## 1. Introduction

The maternal diet before conception and during pregnancy has long-term implications for maternal and offspring health, from placental development [1], risk of developing gestational diabetes [2], birth complications [3], birth weight [4] and risk of developing allergies in childhood [5,6,7,8]. Exposure to an unfavourable environment in early pregnancy is known to significantly increase the risk of diseases in adult life; this is known as ‘early life programming’ and one of the most important factors is the maternal diet [9]. Pregnancy presents an opportune time window for healthcare professionals’ intervention to improve health for both the mother and child, making the evidence surrounding maternal diet an important tool for the healthcare practice. 

Maternal diet is a blueprint for the diet which children are likely to follow into adolescence. This is due to the fact that, typically, mothers are responsible for feeding their children. Without intervention, pregnant women who do not follow a healthy diet or lifestyle choices are unlikely to change their patterns of behaviour. Sometimes obstacles to changes in behaviour can present themselves even for well-known harmful habits such as smoking during pregnancy, but as a unique window of opportunity, focusing health service efforts to impact the behaviour of pregnant women is crucial.

The Mediterranean Diet (MD) and Mediterranean Diet Adherence (MDA) is characterized by a high intake of fruits, vegetables, whole grain cereals, legumes, fish and nuts; low-to-moderate consumption of dairy products and limited amounts of red meat and red wine. It is low in saturated fats and high in antioxidants, fibre and mono and polyunsaturated fatty acids mainly derived from extra virgin olive oil (evoo) (MUFAs) and oily fish (n-3 PUFAs). The MD is known to have many beneficial effects for longevity and disease prevention, demonstrated in numerous high-quality studies, reviews and meta-analyses, making it the most widely studied and evidence-based dietary approach to healthy eating and disease prevention [10,11,12,13]. The unique synergy of various health benefitting nutrients makes MD an effective approach to improving health [11]. 

Recording dietary patterns through food diaries and food frequency questionnaires (FFQs) is a valuable tool to analyse eating behaviours and better understand which foods and food groups are consumed. Individual foods, such as extra virgin olive oil (evoo), as well as food groups, such as pulses, provide an insight to the nutrients consumed in the recorded foods, such as monounsaturated fatty acids (MUFAs) and fibre, respectively. Asking an individual to recount specific nutrient amounts is more complex than recounting what foods were consumed as part of the daily diet, making dietary pattern analysis more attainable for patients and crucial for research.

Evidence on the impact of diet during pregnancy on health outcomes focusses on either individual nutrients, such as folic acid, or the Mediterranean Diet for their impact on specific disease outcomes such as neural tube defects [14] or leukaemia [15]. The degree of evidence available for the impact of specific nutrients in pregnancy on outcomes for offspring health is more powerful than that for dietary patterns. For instance, the necessity for appropriate levels of folic acid during gestation is well-documented in the prevention of neural tube defects, and the mechanism by which folic acid prevents neural tube defects is well understood and documented. Whereas supplements are the quicker and easier way to address certain nutritional deficiencies of individual nutrients [11], dietary patterns are a useful way of studying the effects of dietary exposure, in turn, making recommended diets a potential health improvement intervention alongside appropriate and necessary supplementation in pregnancy. 

The Mediterranean Diet is well established as beneficial in the literature [16]. Previous reviews of the evidence where several dietary patterns have been evaluated highlight that maternal dietary patterns are important factors in early life programming. Maternal dietary patterns which reflect the MD showed consistent associations with a lower risk for allergic disease in children [17], appropriate infant birthweight [18], and lower risk of pre-eclampsia and preterm birth [19]. This Review investigates and presents current evidence exclusively on the Mediterranean Diet’s multifactorial impact on overall health outcomes for both the mother and offspring, as the sole dietary pattern of interest. 

Our results indicate that the Mediterranean Diet has a significantly positive impact on maternal and offspring health, strengthening the evidence base to encourage its adoption as a preventive measure against diseases throughout the life course and especially in pregnancy when the outcome of two or more individuals can be positively impacted. 

## 2. Materials and Methods 

### 2.1. Data Sources and Search Strategy

This systematic review was performed using a predetermined, unpublished protocol and in accordance with standardized reporting guidelines [20]. Search terms identification and classification were guided by previous comprehensive relevant reviews [21]. One reviewer (FA) performed searches in the following online electronic databases (Medline, Embase, Web of Science, Scopus, Maternity & infant care and Cochrane). The search of online databases is up to date to February 2019. The search was not restricted by language or date.

The search was broken down into four main categories. To identify the relevant population, the first Boolean search was done using the term “OR” to explode (search by subject heading) and map (search by keyword) the following MeSH headings “child health” or “offspring” or “newborn” or “neonate” or “child” or “baby” or “gestation” or “pregnancy” or “pregnant woman” or “perinatal period” or “prenatal exposure” or “prenatal”. To identify relevant interventions the second Boolean search used the term “OR” to explode and map “Mediterranean diet” or “fruit” or “vegetable” or “legume” or “nut” or "olive oil” or “evoo” or “oily fish” or “seafood” or “tomato”. The third category of MeSH headings was also related to the intervention and included ((low or little or medium or moderate or less or decrease or reduce or restrict) (intake or consumption or consume or eat or amount)) AND (“dairy product” or “red meat” or “processed meat” or “red wine”). Finally, the fourth group of key terms was used to identify the study design whereby a Boolean search using the term “OR” was used to explode and map the keywords “randomized controlled trial” or “controlled clinical trial” or “placebo” or “Retrospective Studies” or “Cohort Studies”. The four search categories were then combined using the Boolean operator “AND”. The hand searching of results in the Maternity & infant care database for a Mediterranean diet highlighted that using the terminology of ‘child health’ was not capturing some relevant papers, thus, the papers which were the result of searching for a Mediterranean diet in this database were hand-searched instead. For a full list of the search terms in each database please see Appendix A.

### 2.2. Study Selection

Three reviewers (FA, AS and SH) independently evaluated articles for eligibility in a three-stage procedure. In stage one, all identified titles and abstracts were reviewed. In stage two, a full-text review was performed on all the articles that met the predefined inclusion criteria as well as all articles for which there was uncertainty as to eligibility. In stage three, full texts were re-evaluated for data extraction. 

### 2.3. Inclusion/Exclusion Criteria

Studies, publications or reports in the English-language describing an association between the Mediterranean diet during pregnancy and infant development were eligible for inclusion. Studies, publications or reports were included for analysis if they included the following: 

1. Exposure: Comprehensive dietary assessment: Food frequency questionnaire (FFQ), 24 h diet recall, food record, diet history, Use of a priori dietary score or index;

2. Outcome: Clinical neonatal outcome, disorders assessed by study staff, medical records or clinician diagnosed (for example, Foetal growth restriction (FGR) and preterm delivery (PTD)), developmental issues assessed by validated scale/questionnaire; 

3. Design: Observational studies (cross-sectional, cohort, case-control), Interventional studies (clinical trials);

4. Population: Pregnant women (at any gestational age).

Articles were excluded if they investigated pregnant women with other dietary complications (non-gestational diabetes, obesity, anorexia, malnourishment) and/or if they explored a specific nutrient impact as opposed to a dietary pattern (e.g., Omega-3 supplementation as a feature of the Med Diet.)

### 2.4. Data Extraction and Quality Assessment

Two reviewers (FA and AS) independently extracted data from all studies that satisfied the inclusion criteria. Any disagreement in data extraction and/or study inclusion was resolved through discussion between the two reviewers and, when necessary, a third reviewer (SH). 

The primary outcome was the impact on maternal and offspring health. A number of other study characteristics were also extracted including geographic location, description of the study population, primary outcomes, description of intervention and control, and results. Furthermore, data pertaining to sample size, number, and features of the intervention were also extracted.

### 2.5. Quality Assessment

The methodological quality of studies was assessed and scored independently by two reviewers (FA and AS) using the validated National Heart, Lung, and Blood Institute (NHLBI) study Quality Assessment Tool [22] (see Appendix B
Table A1). Disagreements were resolved through discussion. The summary score for each study was calculated (minimum: 1, maximum: 10) and categorised into three categories. Studies with a score of 8 or above were considered of high-quality, studies with a score of 5–7 were considered of fair quality and studies with a score of 4 or below were considered of poor quality. These categories were used to evaluate whether outcomes significantly varied according to study quality and to determine which weight studies should be given in the synthesis of the findings.

### 2.6. Data Synthesis and Presentation 

Since exploring heterogeneity is one of the main aims of an SLR/MA, a descriptive synthesis approach was used instead of meta-analytic procedures. Our review includes studies with diverse designs and all shared comprehensive dietary assessment tools to measure MD adherence, with neonatal and maternal clinical markers as outcomes. The results of the quality assessment are presented in Table 1. As the narrative synthesis was used to analyse and present the findings, the results of data abstraction are summarized in Table 2, which outlines the type, objective and target of intervention, and measurement of effect for each intervention. Study results are summarized in Table 3 according to categories/themes and whether the MD was found to have a Protective, Null or Negative effect on outcomes.

## 3. Results

### 3.1. Literature Selection

Our initial database search is outlined in Figure 1, showing that a total of 125 articles were initially identified based on our search criteria, 14 duplicates were removed, leaving 111 articles for title and abstract screening, of which 67 were excluded for the title (25 were the wrong type of article for this review, 19 targetted the non-objective population, 8 were studies that included dietary complications or risk factors, and 15 focused on a specific nutrient impact); 16 at abstract following the inclusion/exclusion criteria (9 were the wrong type of article for this review, 3 targetted a non-objective population, 3 for studies that included dietary complications or risk factors, and 2 focused on specific nutrient impact); and 6 after full text screening, included in the Appendix C (2 were the wrong type of article for this review, 3 targetted a non-objective population, and 1 focused on specific nutrient impact), for a total of 22 articles reviewed and included in this paper.

The studies included all used a comprehensive dietary assessment tool to measure MD adherence, with neonatal and maternal clinical markers as outcomes. The selected studies included observational, interventional studies, randomized trials, and all studies included a population of pregnant women at any gestational age with neonatal or child follow-ups. Table 1 shows our quality assessment results and Appendix C
Figure A1 shows papers excluded after full text review.

### 3.2. Study Characteristics

The studies included (*n* = 22) were published between 2008 and 2018 and were conducted in 8 countries, which included Spain, USA, Greece, Holland, Norway, Chile, French West Indies, and the United Kingdom. The common dietary measurements used were the adherence to the Mediterranean Diet Score (MDS) and an adapted MDS for pregnancy (MDS-p).

The MDS is an a priori defined score developed to measure compliance to a high intake of Mediterranean Diet foods with a score ranging from 0 to 9, where 9 indicates greater adherence to the diet [13]. The MDS-p is a scoring system that was designed by Montegaudo et al. [45] for pregnant women to better quantify the micronutrients in their diets such as calcium, folic acid, and iron, which are especially relevant in pregnancy [39]. 

Of the included studies, 16 were cohort studies, 1 was a randomized interventional study, 1 was a randomized controlled trial, 2 were case-control studies, and 2 were cross-sectional studies. All studies were conducted during the gestational period with follow-ups after birth. Table 2 shows the summarized characteristics of the studies including design, measures used, and results reported. The total number of participants for all of the studies is 63,336.

### 3.3. MAIN RESULTS

#### 3.3.1. Allergic Disorders

Four studies focused on the relationship between maternal adherence to a Mediterranean Diet during pregnancy and neonatal or childhood outcomes of asthma and atopy. A cohort study by Chatzi, L; Torrent, M.; et al. [27] found that adherence to the MD during pregnancy supports a protective effect against asthma-like symptoms and atopy in childhood at 6.5 years old. Chatzi, L.; Garcia, R.; et al. [28] found that while adherence to the MD was not associated with the risk of wheeze and eczema in any cohort, high meat intake during pregnancy might increase the risk of wheeze during the first year of life. Through questionnaires of epidemiological factors, maternal diet during pregnancy, and childhood diet, Castro-Rodriguez, J.; M. Ramirez-Hernandez et al. [36], found that wheezing, rhinitis or dermatitis were negatively affected by high potato and pasta consumption by the mother and subsequent low fruit and high meat consumption by the child. Lange, N. et al. [37] however, examined 1376 mother-infant pairs and found that the dietary pattern is not associated with recurrent wheeze. 

#### 3.3.2. Premature Birth, Birth Weight, Childhood Obesity

Six studies examined the implications of the diet on premature birth, gestational diabetes (please see Table 3), birth weight, and/or childhood obesity. Parlapani, E. et al. [29] found that neonates of mothers with low adherence to the MD had significantly higher intrauterine growth restriction and lower birth weights. Haugen, M. et al. [34] investigated the association between women who met the MD adherence criteria and those who did not. The study showed that those who met the MD criteria of fish ≥2 times a week, fruit and vegetables ≥5 times a day, use of olive/canola oil, red meat intake ≤2 times a week, and ≤2 cups of coffee a day, did not have a lower risk of preterm birth compared to women who met none of the criteria. A population-based cohort study involving 922 late and moderate preterm births (LMPT) by Smith, L. et al. [44], identified that women who did not include any aspects of the MD during pregnancy were nearly twice as likely to have LMPT in comparison to higher adherence mothers. Fernandez-Barres, et al. [30] evaluated associates between adherence to the MD during pregnancy, but found no risk to childhood obesity, however, it was associated with a lower waist circumference, which is related to abdominal obesity and is an important health marker. Peraita-Costa et al. [40] administered the “Kidmed” questionnaire to collect dietary information from mothers on adherence to the MD. Of the 492 women, 40.2% showed low adherence to the MD, but this study did not directly correlate the results to low birth weight due to potentially confounding factors including smoking, low education levels, and low dairy intake. Saunders, L. et al. [41] found that low adherence to the MD could implicate a positive correlation between maternal diet and small birth weight. 

#### 3.3.3. Cardiometabolic and Congenital Defects

Three studies examined the link between better maternal diet quality with congenital heart defects or metabolic factors. Chatzi, L.; Rifas-Shiman, S.; et al. [26] studied 997 mother-child pairs, finding that improved adherence to the MD during pregnancy showed lower systolic and diastolic blood pressures and may protect offspring against cardiometabolic risk. Botto, L. [25] et al. found, in a high-quality study, that diet quality was a factor of reduced conotruncal and atrial septal heart defects. Vujkovic, M. et al. [43] found that because of the higher levels of serum and RBC folate, Vitamin B12 and lower plasma homocysteine contained in the MD, offspring have a reduced risk of spina bifida.

#### 3.3.4. Gestational Diabetes and Pre-Eclampsia

Two studies found that gestational nutrition based on the MD reduces the incidence of Gestational Diabetes (GDM). In a prospective randomized interventional study by A Duran, [24] 177 women out of 874 were diagnosed with GDM and found that with early intervention using the MD, gestational outcomes were improved. Likewise, the results of a randomized controlled trial conducted by Assaf-Balut, C. et al. [24] show that the incidence of GDM was reduced by early dietary intervention with a diet rich in EVOO and pistachios. Parlapani et al. [29] found that MDA was an independent predictor of gestational hypertension and pre-eclampsia. 

#### 3.3.5. DNA Methylation

Two studies examined the association between maternal adherence to the MD and DNA methylation in infants. One study conducted by Gonzalez-Nahm et al. [33] showed evidence that maternal diet of low MD adherence had an increased risk of female sex-linked hypo-methylation at the MEG3-IG differentially methylated region. House, J. et al. [35] also found an association between maternal adherence to the MD and female sex-linked methylation at MEG3, IGF2, and SGCE/PEG10 DMRs, and identified an association between MD and favourable neurobehavioral outcomes in early childhood.

#### 3.3.6. Biomarkers

Four studies were associated with adherence to the MD and subsequent repercussions on biomarkers and implications of the pregnancy and on the newborn. Gesteiro, E. et al. [31] aimed to determine the relationship between diet within the first trimester and biomarkers of insulin resistance at birth. They found that women consuming low MD adherence diets had low-fasting glycaemia and delivered infants with high insulinaemia. A follow up cross-sectional study by Gasteiro, E., Bastida, S. et al. [32] aiming to identify the relationship between diet quality during pregnancy and serum lipid, arylesterase and homocysteine values at birth identified that neonates whose mothers had low adherence to the MD presented impaired levels of the referenced biomarkers. In a cohort study by Mantzoros et al. [38], multivariable linear regression was used to analyse the correlation between maternal diet during the 1st and 2nd trimesters and cord blood levels of leptin and adiponectin. High adherence to the MD was not found to be associated with these levels. Conversely, a study by Monteagudo et al. [39] on the exposure to organochlorine pesticides and maternal diet indicated that higher folic acid supplementation and greater exposure to the endocrine-disrupting residues were related to higher newborn weight.

#### 3.3.7. Behavioural Development

One cohort study by Steenweg-de Graaff et al. [42], examining the pregnancies and follow-up of the 3104 children born, found that the MD was not associated with internalizing problems such as anxiety or depression, while low adherence to the MD was positively associated with increased child externalizing problems, such as aggression or inattention.

Of the 22 studies included, 18 found that adherence to the Mediterranean Diet during pregnancy had protective factors on the health of the newborn and 4 of the studies were inconclusive or showed no correlation. None of the studies showed a negative association between the MD and the outcomes. 

## 4. Discussion

The mechanisms by which the MD exerts its effect on fetal development and maternal health are complex and need further research. By reviewing the available literature focusing specifically on MD interventions/exposure, as opposed to all dietary patterns as in previous reviews, we have a good overview of how the MD pattern effects mothers and their offspring on a variety of outcomes. Particularly interesting insights can be gleaned from the numerical values associated with some of the risk factors. The studies in this review are of good quality and represent a combined study population >63,000. 

The observed heterogeneity of interventions, study populations and outcomes measured, allows for a big picture view of how the MD diet exerts its effects at all gestational ages and in the early years of life. For instance, when looking at behavioural outcomes, this review concluded that adherence to the MD in pregnancy has a statistically significant impact on decreasing the likelihood of offspring exhibiting depressive behaviours (OR 0.28) [35] in an ethnically diverse cohort, and another study added to this with a reduced OR of 0.90 [42] for developing externalising behaviours (such as aggression) for offspring of mothers with a high MDA. 

Though some of the outcome categories we have identified only have 2 or 3 studies, they give a good starting point for future research to build on the current knowledge. Berti et al.’s comprehensive literature review of early life nutritional exposures on life-long health splits its findings into ‘sections’ [46] which, again, give a big picture overview on the vast impact that maternal, early life and even pre-conceptional nutrition can have. Unlike some more focused reviews, investigating the impact of maternal diet on allergic diseases [47], for instance, we hope that our review’s heterogeneity contributes to a wider understanding of maternal dietary impact. 

This review brings together some ideas of how the MD may be contributing to in-utero development through specific mechanistic pathways. The methylation of specific sites in two of the studies (see Table 3) present results on how the MD, as opposed to specific nutrients, contribute to this very time-sensitive window of change of DNA methylation during fetal development. 

The results on atopy, asthma and eczema highlight that whilst the MD is not always protective, it is associated with a reduced likelihood of allergic disorders as a result of maternal MD diet adherence and compounded further by offspring MD adherence. This type of follow up warrants attention; further investigation into the impacts which MD adherence for offspring of mothers who had high adherence to the MD during pregnancy and lactation, compared to non-MD adherent mother-child pairs would allow for dietary recommendations to be evidence-based through the course of pregnancy and early life as a one-time course. Intuitively, this would be a good way to approach dietary recommendations for improving health, as most children adopt their mother’s dietary choices as the main provider of food in early life. 

With regards to premature birth, low birth weight and childhood obesity, the evidence is more mixed. The studies that found no association between the MD and decreased likelihood of premature birth were of high quality. Some of the uncertainty is due to the limitations of the studies, some not controlling for confounders of specific outcomes like intrauterine growth, such as smoking. Despite this, the studies that were inconclusive on the impact of the MD on their primary chosen outcome, for instance the INMA birth cohort study for childhood obesity [30] and Parlapani et al.’s work [29] on birth complications and prematurity, still found statistically significant positive associations between MD adherence and other health outcomes in their results, such as waist circumference and likelihood of pre-eclampsia and necrotizing enterocolitis for each study, respectively.

A strong case for the positive impact of the MD can be seen in the results of the studies focusing on cardiometabolic and congenital defects, GDM and DNA methylation. Though these studies investigated hugely different markers and outcomes, it is clear that the MD has a protective effect on them all. The evidence presented here regarding DNA methylation and biomarkers reflects a novel way to measure the impact of dietary changes and contributes to the understanding of the mechanisms behind the long-term impacts which ‘early-life programming’ has. The nature of DNA methylation in the fetal development time-course makes it possible to observe the impact which specific intra-uterine exposures have on offspring DNA methylations 20, 30 or 40 years later. For example, studies on the impact of intra-uterine exposure to famine showed that lower methylation of 5 CpG dinucleotides within the insulin-like growth factor-II differentially methylated region (DMR) [48] was detected in affected offspring decades later. Thus, the potential for future research on the impact of the MD on methylation sites of interest is vast, as dietary exposure has such a large impact. 

The pattern of MD that may influence the reduction of adverse effects most appears to be one where there is purposefully added evoo and nuts. As seen in the present literature, increased vegetables and oily fish and decreased processed foods delivered positive results, and the effects are seen both in metabolically healthy and compromised gestation, indicating that the MD could be a diet with beneficial outcomes for GDM and metabolically healthy pregnancies. 

The results of this review add to the growing body of evidence that the Mediterranean Diet is a beneficial dietary recommendation throughout the life course. By collating the evidence on outcomes of the MD for the mother and child, we highlight the broad range of effects this dietary intervention has. The limitations of this work are the small number of studies per outcome group, with the biggest group here being ‘premature birth and birth weight’ at only 7 studies total. Regular updates of this review are important as research around the topic is growing, and with an expanding evidence base, the possibility of meta-analysis for each category is likely. The biggest strength of this review is that, even with small numbers of studies, it highlights the impact which MDA in pregnancy and early childhood can have on several different health outcomes. The implications for clinical practice are great, as prescribing an MD pattern to women of reproductive age is a simple intervention, with important clinical potential for both the mother and offspring. 

## 5. Conclusions

The Mediterranean Diet in pregnancy and early infancy is safe and beneficial for a wide range of maternal and offspring outcomes. Further research to ascertain the relationship and mechanisms maternal MD has with health outcomes of interest in different populations is needed to position it as a public health intervention for all populations. 

## Figures and Tables

**Figure 1 nutrients-11-01098-f001:**
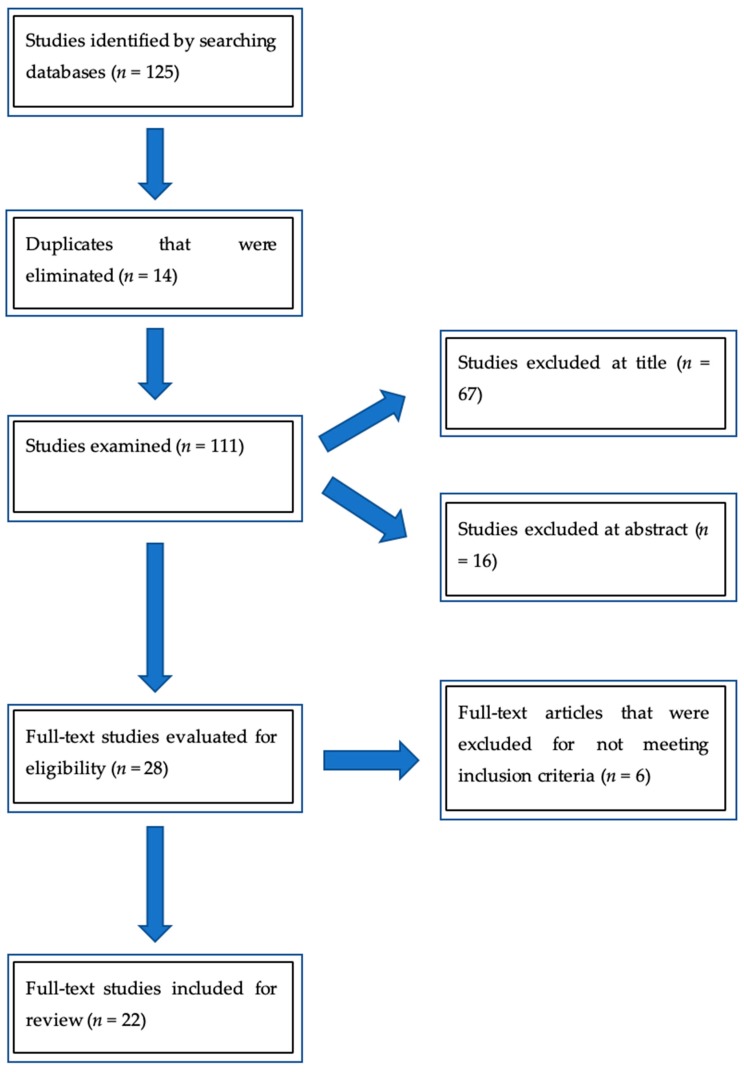
The flow diagram of the process for study selection.

**Table 1 nutrients-11-01098-t001:** The quality assessment score values: <4 = Low, 4–7 = Medium, 8–10 = High.

Author	Design	Score (1–10)	Quality Assessment
Assaf-Balut, C; Garcia de la Torre, N; A Duran et al. [23]	Prospective randomized interventional study	8	High
Assaf-Balut, Carla; Garcia de la Torre, et al. [24]	Prospective randomized controlled trial (The St Carlos GDM prevention study)	9	High
Botto, Lorenzo D.; Krikov, Sergey; et al. [25]	Multicentre population-based case-control study	9	High
Chatzi, L.; Rifas-Shiman, S.; [26]	Prospective Cohort study (Project Viva + Rhea Study)	9	High
Chatzi L.; Torrent, M.; et al. [27]	Prospective Cohort study	7	High
Chatzi, L.; Garcia, R.; et. al. [28]	Cohort study (INMA and Rhea Study)	9	High
E, Parlapani; et al. [29]	Cohort study	7	Medium
Fernandez-Barres, S.; et al. [30]	Birth Cohort study (INMA)	8	High
Gesteiro, E.; Rodriguez B., et al. [31]	Cohort study	7	Medium
Gesteiro, E.; Bastida S., et al. [32]	Cross-sectional study	7	Medium
Gonzalez-Nahm, S. et al. [33]	Cohort study	8	High
Haugen, M.; et al. [34]	Prospective Cohort study (MoBa)	8	High
House, J.; et al. [35]	Prospective Cohort study	9	High
Castro-Rodriguez, J.; et al. [36]	Cohort study	7	Medium
Lange, N. [37]	Longitudinal pre-birth cohort study	9	High
Mantzoros, C.; et al. [38]	Prospective cohort study (Project Viva)	8	High
Monteagudo, C.; et al. [39]	Cohort study	8	High
Peraita-Costa, I. et al. [40]	Retrospective cross-sectional population-based study	7	Medium
Saunders, L.; et al. [41]	Cohort study (TIMOUN)	8	High
Steenweg-de Graaff, J.; et al. [42]	Population-based cohort (The Generation R Study)	9	High
Vujkovic, M.; et al. [43]	Case-control study	7	Medium
Smith, L.; et al. [44]	Population-based cohort study	7	Medium

GDM = gestational diabetes mellitus, INMA = INfancia y Medio Ambiente study, TIMOUN= French Caribeean Mother Child cohort study.

**Table 2 nutrients-11-01098-t002:** The outcomes and effect estimates for the included studies.

Authors	Design and Cohort	Included Participants and Gestational Age	Intervention Type and Comparator	Results
Assaf-Balut, C; Garcia de la Torre, N; A Duran et al. [23] Spain	Prospective randomized interventional study	874; First Trimester	MD nutritional therapy	As an early nutritional intervention, MD reduces the incidence of GDM. Comparison of HbA_1c_ levels at 24–28 weeks in women with GDM and normal glucose tolerance: *p* = 0.001. Values became similar at 36–38 gestational weeks with intervention.
Assaf-Balut, Carla; Garcia de la Torre, et al. [24] Spain	Prospective randomized controlled trial	874; intervention group (IG), *n* = 434 control group (CG), *n* = 440; 8–12 gestational weeks (First Trimester)	MD nutritional therapy with additional evoo and pistachios	Supplemented MD reduces the incidence of GDM as an early nutritional intervention. IG showed reduced rates of insulin-treated GDM: *p* =< 0.05
Botto, Lorenzo D.; Krikov, Sergey; et al. [25] USA	Multicentre population-based case-control study	Mothers of babies with major non-syndromic congenital heart defects (*n* = 9885); mothers with unaffected babies (*n* = 9468); maternal diet assessed in the year before pregnancy	A priori defined MDS with Quartiles 1–4 (worst to best)	Better diet quality is associated with a reduced occurrence of some conotruncal and septal heart defects Overall conotruncal defects: OR 0.63, 95% CI 0.49 to 0.80 Overall tetralogy of Fallot: OR 0.76, 95% CI 0.64 to 0.91 Overall septal defects: OR 0.77, 95% CI 0.63 to 0.94 Overall atrial septal defects: OR 0.86, 95% CI 0.75 to 1.00
Chatzi, L.; Rifas-Shiman, S.; [26] USA, Greece	Cohort study; Project Viva	Mother-child pairs from USA: 997 Greece: 569 MDA measured during pregnancy with follow-up at median 4.2 and 7.7 years	MDA with a priori defined MDS through FFQ	Greater adherence to MD during pregnancy may protect against excess offspring cardiometabolic risk. For each 3-point increase in MDS, offspring BMI decreased by 0.14 units (95% confidence interval, −0.15 to −0.13)
Chatzi L.; Torrent, M.; et al. [27] Spain	Cohort study	507 mothers during the gestational period; 460 children at 6.5 years post-gestational follow-up	Impact of MDA during pregnancy on asthma and atopy in childhood using a priori defined MDS	Adherence to Med Diet during pregnancy support protective effect against asthma-like symptoms and atopy in childhood Persistent wheeze: OR 0.22; 95% CI 0.08 to 0.90 Atopic wheeze: OR 0.30; 95% CI 0.10 to 0.90 Atopy: (OR 0.55; 95% CI 0.31 to 0.97
Chatzi, L.; Garcia, R.; et al. [28] Spain, Greece	Cohort study; INMA (Spain) RHEA (Greece)	During pregnancy with follow-up within 1 year post-gestational; 1771 mother-newborn pairs (Spain); 745 pairs (Greece)	MDA calculated through completed FFQ	High meat intake during pregnancy may increase the risk of a wheeze in the first year of life; high dairy intake may decrease it RR 0.83, 95% CI 0.72, 0.96
E, Parlapani; et al. [29] Greece	Cohort study	82 women delivering preterm singletons ≤34 weeks	FFQ and MDA	High adherence to MD, may favourably affect intrauterine growth (IUGR), premature birth and maternal hypertension (HTN); Low-MDA neonates group had a higher rate of IUGR: OR 3.3 Low-MDA mothers had a higher rate of prematurity: OR 1.6 Low-MDA mothers had a higher gestational HTN: OR 3.8
Fernandez-Barres, S.; et al. [30] Spain	Cohort study; INMA	1827 mother-child pairs, assessed during pregnancy	FFQ and MDA	Adherence to MD during pregnancy not associated with a risk of childhood obesity, but is linked to a lower waist circumference; *p*-value for trend = 0.009
Gesteiro, E.; Rodriguez B., et al. [31] Spain	Cohort study	35 women with ‘adequate’ or ‘inadequate’ diets according to HEI (healthy eating index) and MDA score; 1st trimester	13 point MDA score via FFQ	Maternal diets during the 1st trimester with low HEIs or adherence to MD have a negative effect on insulin markers at birth; Low MDA-score diets had low-fasting glycaemia: *p* = 0.025 and delivered infants with high insulinaemia: *p* = 0.049
Gesteiro, E.; Sanchez-Muiz FJ, et al. [32] Spain	Cross-sectional study	53 mother-neonate pairs; GDM screening at 24–28 gestational weeks	Maternal MDA and offspring lipoprotein profile	Neonates of mothers who consumed low adherence of MD during pregnancy presented impaired lipoprotein and higher homocysteine levels; Mothers’ diet in the nAA + AT x mTT group (neonates carrying FTO rs9939609 T allele x Mothers homozygous for FTO rs9939609 T allele) had a significantly lower MDA score: *p* = 0.05
Gonzalez-Nahm, S. et al. [33] USA	Cohort study	390 women whose infants had DNA methylation cord blood data available; FFQ at preconception or 1st trimester	MDA via FFQ	Suggests that maternal diet can have a sex-specific impact on infant DNA methylation at specific imprinted DMRs; OR = 7.40, 95% CI = 1.88–20.09
Haugen, M.; et al. [34] Norway	Cohort study; MoBa	MD criteria met: 569 women; 1–4 criteria met: 25,397 women; 0 MD criteria met: 159 women; 18–24 gestational weeks	MDA via FFQ	Women who adhered to the MD criteria did not have a reduced risk of preterm birth compared to women who met none of the criteria; OR: 0.73, 95% CI: 0.32, 1.68 Intake of fish twice a week or more associated with lower preterm birth; OR: 0.84; 95% CI: 0.74, 0.95
House, J.; et al. [35] USA	Cohort study; NEST	325 mother-infant pairs; 1st trimester; follow-up at 2 years post-gestation	MDA via FFQ	Offspring of women with high MDA less likely to exhibit neurobehavioural effects: Depression; OR = 0.28 Anxiety; OR = 0.42 Social relatedness; OR = 2.38
Castro-Rodriguez, J.; et al. [36] Spain, Chile	Cohort study	Gestational period; follow-up in 1000 preschoolers (at 1.5 yrs and 4 yrs)	MDA via FFQ	Low fruit and high meat consumption by the child had a negative effect on allergic responses (wheezing, rhinitis, or dermatitis); as did the high consumption of pasta and potatoes by the mother
Lange, N. [37] USA	Longitudinal prebirth cohort study; Project Viva	1376 mother-infant pairs; 1st and 2nd trimesters with follow-up at 3 years post-gestation	MDA via FFQ	Dietary pattern during pregnancy not associated with recurrent wheeze; OR per 1-point increase in MD: 0.98, 95% CI, 0.98–1.08
Mantzoros, C.; et al. [38] USA	Prospective cohort study; Project Viva	780 women; 1st and 2nd trimesters; post-gestational cord blood	MDA	Adherence to MD during pregnancy not associated with cord blood leptin or adiponectin; *p*-value = 0.38
Monteagudo, C.; et al. [39] Spain	Cohort study	320 umbilical cord serum samples	MDS-p (med diet score adapted to pregnancy)	Adherence to the MD and folic acid supplementation during pregnancy may indicate being overweight in newborns; OR = 3.33 (*p* = 0.019)
Peraita-Costa, I. et al. [40] Spain	Retrospective cross-sectional population-based study	492 mother-child pairs; immediately post-delivery and for 6 months thereafter	MDA with two groups identified: low and high adherence	Low adherence to an MD was not associated with a higher risk of a low birthweight newborn; aOR = 1.68; 95% CI 1.02–5.46
Saunders, L.; et al. [41] French West Indies	Cohort study; TIMOUN	728 pregnant women who delivered liveborn singletons with no malformations	Semi-quantitative FFQ analysed for MDA	Results suggest that adherence to a Caribbean diet may include benefits of MD, contributing to a reduction in preterm delivery in overweight women; A OR: 0.7, 95% CI 0.6, 0.9
Steenweg-de Graaff, J.; et al. [42] The Netherlands	Population-based cohort; Generation R Study	During pregnancy at median 13.5 weeks; Post-gestation in 3104 children at 1.5, 3, and 6 years of age	MDA via FFQ	High adherence to traditional Dutch diet and low adherence to MD are linked to an increased risk of child externalizing problems; OR per SD in MDS: 0.90, 95% CI: 0.83–0.97 OR per SD in Traditionally Dutch Score: 1.11, 95% CI: 1.03–1.21
Vujkovic, M.; et al. [43] The Netherlands	Case-control study	50 mothers of children with Spinal Bifida; 81 control mothers post-gestation	Dietary assessment via FFQ	MD seems to show an association with reducing the risk of offspring being affected by SB; Weak MDA; OR: 2.7 (95% CI 1.2–6.1) High MDA; OR: 3.5 (95% CI 1.5–7.9)
Smith, L.; et al. [44] United Kingdom	Population-based cohort study	922 LMPT; 965 term births; 32–36 weeks gestation (3rd Trimester)	Maternal interview for dietary factors: MDA, low fruit and vegetable intake, use of folic acid supplements	Women with 0 adherence to MD were nearly twice as likely to deliver LMPT; RR 1.81 (1.04 to 3.14) Smokers and low consumption of fruit and vegetables had a particularly high risk; RR 1.81 (1.29 to 2.55)

MD = Mediterranean Diet, MDS = Mediterranean Diet Score, MDA = Mediterranean Diet Adherance, FFQ = Food Frequency Questionnaire, HEI = Healthy Eating Index.

**Table 3 nutrients-11-01098-t003:** A summary of findings from our review of the evidence.

Health Outcomes	Impact of Maternal MD
Outcome Variable	Protective	Negative	Inconclusive
Allergic Disorders	3 (27,28,36)		1 (37)
Premature birth, birth weight, childhood obesity	5 (29,30,39,41,44)		2 (30,36)
Cardiometabolic and congenital defects	3 (25,26,43)		
Gestational diabetes & pre-eclampsia	2 (23,24,29)		
DNA Methylation	2 (33,35)		
Biomarkers	2 (31,32)		1(38)
Behavioural development	1 (42)

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
