# Peer review of "The Impact of Mediterranean Dietary Patterns During Pregnancy on Maternal and Offspring Health"

_nutrients, 2019, doi:10.3390/nu11051098_

Round 1

Reviewer 1 Report

The authors study the effects of the Mediterranean diet on maternal and fetal and postnatal health. It is a good systematic review.

1.       Perhaps, in table 2, they should show the Mediterranean diet comparator used in each study. In nutrition, as important is what you eat as it is instead of eating it. This can modulate the success of the intervention with Mediterranean Diet.

2.        Additionally, in the discussion they could comment on which pattern of the Mediterranean diet may influence more in the reduction of maternal-fetal adverse events, as recently published ( nutrients 2019, 11, 66; doi:10.3390/nu11010066) and the effects of the Mediterranean diet in metabolically healthy gestation (without GDM) (Ann Nutr Metab 2019;74:69–79)

Author Response

Thank you for taking the time to review our manuscript.

(1) - Table 2 now clearly has a 'comparator used' column. 

(2) - Line 363 adds some detail on which patterns are highlighted in our review as being most beneficial. Thank you for bringing this to our attention. 

We hope this satisfies the reviewer's remarks and we thank them for their positive review of our work.

Best,

FA

Reviewer 2 Report

The manuscript addresses an interesting and timely subject of the impact of Mediterranean dietary patterns during pregnancy on maternal and offspring health.

Please consider the following:

1.     The method of systematic review is not powerful and deterministic enough to establish whether following a Mediterranean diet in pregnancy is beneficial for maternal and offspring outcomes. Please use less powerful words during the weight of evidence provided. I suggest changing ¨establish¨ in line13 with ¨assessing¨. Also, the word ¨found¨ in line 316 with the word ¨concluded¨.

2.     Please revise the references number to accurate reflect each mention. For example No 26 and No 45 do not correspond to the references mentioned in the text (lines 214 and 216).

Author Response

Thank you for taking the time to review our manuscript so thoroughly. We agree that the ideal methodology would have included a meta-analysis had it been possible to do one, however we are glad that you find our research interesting and timely.

(1) We appreciate that the wording used in the highlighted points were too deterministic and have changed both. 

(2) We have revised the references, thank you for pointing these out, there was a glitch with the software used (Zotero) but hope this is now resolved. 

Best Wishes,

FA

Reviewer 3 Report

This is a systematic review of studies aimed at assessing the impact of Mediterranean diet consumption during pregnancy on maternal and offspring health.  This is an impressive topic, that is not easy to counter, due to several reasons.  At first, the adherence to Mediterranean diet is probably different between people living in the Mediterranean area and those living in western countries and the comparison between the respective dietary patterns sounds complicated. Moreover, the topic reported in the title “maternal and offspring health” seems excessively broad to be addressed in a single systematic review.  On the other hand, some systematic review and meta-analyses are already available on the relationship between Mediterranean diet in pregnancy and allergy and atopic disease, or gestational diabetes. In fact, the Authors recognize the heterogeneity of interventions, study populations and outcomes measured. Although showing a big picture view of the possible effects of the Mediterranean diet in pregnancy, the results do not allow to support other benefits than those already described in other systematic reviews.

I suggest to focus the systematic review on the relationship between Mediterranean diet and   prevention of specific diseases more than on different conditions.

Introduction

Line 53-55

Nutrition epidemiology is based on FFQ and food diaries, both collecting information on food consumption (not nutrients) and not on asking an individual to recount specific nutrient amounts

Line 57-58

The degree of evidence is quite different for nutrient intake and prevention of specific diseases (for example folic acid and neural tube defects or iron and anemia) in comparison with dietary habits (like as Mediterranean diet) and, as an example, cardiometabolic prevention. This should be explained in detail.

Author Response

Thank you for taking the time to review our manuscript. We appreciate that the topic is broad and perhaps covers more topics than normally found in one systematic review, however, we find it gives an important overview of the MD as a factor in maternal and offspring health worth presenting. 

By taking the different outcomes into consideration together, we are able to offer readers a more holistic picture than the existing reviews on individual health outcomes, for e.g. atopic eczema. Though more specific reviews of disease outcomes are important and needed, we feel this review adds a different dimension to existing work, and one that is useful in practice too. 

We have addressed all of the individual comments for working and explanation, specifically expanding lines 61-67 to address the concerns raised for the difference in degree of evidence available.

We hope you will find these changes satisfactory and the work of interest for the Nutrients readership.

Kind Regards,

FA

Round 2

Reviewer 3 Report

In this version the manuscript could be considered for publication.